# Cavitation Effect in Ultrasonic-Assisted Electrolytic In-Process Dressing Grinding of Nanocomposite Ceramics

**DOI:** 10.3390/ma14195611

**Published:** 2021-09-27

**Authors:** Guangxi Li, Fan Chen, Wenbo Bie, Bo Zhao, Zongxia Fu, Xiaobo Wang

**Affiliations:** 1School of Electrical and Mechanical Engineering, Pingdingshan University, Pingdingshan 467000, China; lgxleaning@163.com (G.L.); wenbo187120@163.com (W.B.); fuzongxia@126.com (Z.F.); 2School of Mechanical and Power Engineering, Henan Polytechnic University, Jiaozuo 454000, China; zhaob@hpu.edu.cn (B.Z.); wangxb@hpu.edu.cn (X.W.)

**Keywords:** nanocomposite ceramics, cavitation effect, oxide film, surface quality, ultrasonic-assisted ELID grinding

## Abstract

Ultrasonic-assisted electrolytic in-process dressing (UA-ELID) grinding is a promising technology that uses a metal-bonded diamond grinding wheel to achieve a mirror surface finish on hard and brittle materials. In this paper, the UA-ELID grinding was applied to nanocomposite ceramic for investigating the cavitation effect on the processing performance. Firstly, the ultrasonic cavitation theory was utilized to define the cavitation threshold, collapse of cavitation bubbles, and variation of their radii. Next, the online monitoring system was designed to observe the ultrasonic cavitation under different ultrasonic amplitude for the actual UA-ELID grinding test. A strong effect of ultrasonic cavitation on the grinding wheel surface and the formed oxide film was experimentally proved. Besides, under the action of ultrasonic vibration, the dressing effect of the grinding wheel was improved, and the sharpness of grain increased by 43.2%, and the grain distribution was dramatically changed with the increase of ultrasonic amplitude. Compared with the conventional ELID (C-ELID) grinding, the average protrusion height increased by 14.2%, while the average grain spacing dropped by 21.2%. The UA-ELID grinding reduced the workpiece surface roughness *R_z_* and *R_a_* by 54.2% and 46.5%, respectively, and increased the surface residual compressive stress by 44.5%. The surface morphology observation revealed a change in the material removal mechanism and improvement of the surface quality by ultrasonic cavitation effect. These findings are considered instrumental in theoretical and experimental substantiation of the optimal UA-ELID grinding parameters for the processing of nanocomposite ceramics.

## 1. Introduction

Alumina (Al_2_O_3_) ceramics, as an advanced material with high strength, hardness, and wear/corrosion resistance, has been extensively used in heavy-duty mechanical parts and structures [1]. However, its poor impact resistance and fracture toughness restrict its applicability to dynamically loaded structures and vehicles. With the development of nanotechnology, this limitation was mitigated by adding the second-phase (e.g., ZrO_2_) nanoparticles to the Al_2_O_3_ matrix, which produced the Al_2_O_3_ composite ceramics [2]. The ZrO_2_-toughened Al_2_O_3_ nano-ceramics, also referred to as ZTA ceramics, belong to the hard, brittle material, which is difficult to process via conventional methods [3]. Therefore, alternative processing technologies have been introduced, such as the elastic emission machining (EMM) [4], pre-stressed processing [5], rotary ultrasonic machining (RUM) [6], ELID ultra-precision grinding [7], magnetic polishing technology [8], ultrasonic-assisted grinding [9], and UA-ELID grinding [10]. The latter method is a compound processing through superimposing ultrasonic vibration on the C-ELID grinding, in which the grinding wheel could achieve online dressing, and ensure the enhanced processing efficiency and surface quality [11]. The ultrasonic vibration combination with the C-ELID grinding was found to generate the ultrasonic cavitation effect, which was a unique physical process and a complex nonlinear acoustic phenomenon induced by ultrasonic vibration in liquids [12]. During the cavitation process, a series of dynamic processes take place with the tiny bubbles generation, expansion, closure and collapse [13]. During the UA-ELID grinding, the tens of thousands of micro-voids form cavitation bubbles, which grow and close in the negative and positive pressure zones, respectively, and finally collapse with the help of the alternating positive and negative pressures. At the moment of the cavitation bubble collapse, a high local instantaneous pressure jump occurs and generates a micro-jet with a high velocity, which exerts stronger impact on the processed surface [14].

Many scholars have carried out researches on cavitation mechanism and its application in the EDM, material deburring, ultrasonic honing, micro-drilling, and other fields. Liew et al. [15] adopted ultrasonic cavitation-assisted micro-EDM in micro-holes and found that the oscillation of cavitation bubble group could significantly improve the discharge rate of debris, and decline the debris adhesion. Liang et al. [16] investigated the effect of ultrasonic cavitation on the micro drilling stainless steel and reported that it substantially improved the adhesion and winding of cuttings to the tool and enhanced the quality and accuracy of micro-holes. Toh [17] studied the ultrasonic cavitation effect on the milling stainless steel and concluded that it could effectively decrease the burr height. During the chemical mechanical polishing sapphire, Xu et al. [18] observed that the cavitation effect produced a strong blasting pressure, which repeated action on the surface enhanced the material removal rate (MRR) and declined the surface flatness error to a certain extent. Ye et al. [19] analyzed the impact characteristics of micro-jet from the cavitation bubble collapse in ultrasonic honing using the coupled approach of smoothed-particle hydrodynamics and finite element method (SPH-FEM) and identified the optimal parameters that could improve the surface quality. The above survey proves that the ultrasonic cavitation has a significant effect on the MRR and surface quality.

The UA-ELID grinding processes imply their multidisciplinary consideration: during this processing, the ultrasonic power generates the high-frequency oscillation signal, which is then converted into a high-frequency mechanical vibration by the transducer, and finally, it is transmitted to the grinding wheel by the horn. The motion path of grains will be affected by ultrasonic vibration, and its kinematics are changed under the coupled effect of high vibration and online dressing. As far as the UA-ELID grinding nanocomposite ceramics are concerned, Liu et al. [20] established the model of MRR for hard brittle materials. It was concluded that the MRR increased with an increase of grinding depth at the lower speed of grinding wheel. Shao et al. [21] studied the characteristics of the oxide film in UA-ELID grinding of nanoceramics and analyzed its effect on machining quality and efficiency. Zhao et al. [22] conducted a series of tests and found that during the UA-ELID grinding, the grinding force was reduced by 60%, the MRR increased by 70%, and the surface roughness dropped by 10%, as compared to the respective parameters of the C-ELID grinding. However, to the best of the authors’ knowledge, the ultrasonic cavitation effect on the characteristics of grinding wheel and surface quality of the workpiece during UA-ELID grinding of nanocomposite ceramics had been yet scarcely studied and reported.

Therefore, in this paper, the cavitation effect during UA-ELID grinding was studied to analyze its influence on the machining process. Firstly, the mechanism of ultrasonic cavitation was analyzed, and the cavitation threshold and collapse of the cavitation bubble in UA-ELID grinding were determined. The variation of the cavitation bubble radius in the grinding zone was obtained as well. Secondly, the ultrasonic cavitation under different ultrasonic amplitude was observed through online monitoring. The surface morphology of the grinding wheel under the ultrasonic cavitation was analyzed. Finally, the cavitation effect on the grinding wheel characteristics and the machined surface quality was discussed, in order to reveal the material removal mechanism in the UA-ELID grinding process.

## 2. Mechanism of Ultrasonic Cavitation during UA-ELID Grinding

### 2.1. Principle of UA-ELID Grinding

The schematic of UA-ELID grinding is depicted in Figure 1. The system mainly contains an ultrasonic vibration system, ELID power, ultrasonic cathode device, and diamond grinding wheel with cast iron bond. During the grinding, the diamond grinding wheel is selected as the anode and the copper cathode as the cathode. The ultrasonic vibration is superimposed on the grinding wheel along its axial direction. The electrolytic grinding liquid is persistently supplied during the inter-electrode gaps through an electrolyte nozzle. The UA-ELID grinding principle can be subdivided into four stages; namely, (i) grinding wheel shaping, (ii) grinding wheel pre-dressing to protrude the diamond grains on the wheel surface, (iii) dynamic grinding, and (iv) polishing. Before the processing, the dressing device of electrical discharge machining (EDM) is used to pre-dress the grinding wheel, in which the process of the bonding material (iron) is gradually corroded by the electrochemical reaction, and an oxide film is generated on the surface of the grinding wheel. When its thickness reaches a critical value, the electrochemical reaction is nearly terminated, and the oxide film thickness is no longer increased. Under the ultrasonic vibration effect, the grinding wheel vibrates, scratching the oxide film and the electrochemical reaction continues until a new balanced state is reached. In this study, the third stage (dynamic grinding) is explored to observe the cavitation effect on the processing.

When the ultrasonic wave propagates in the liquid, the vibration of medium molecules can occur. The average distance between molecules will periodically decrease and increase in the compression phase and sparse phases, respectively. If the negative pressure of liquid under the action of ultrasonic vibration is large enough to enhance the average distance between neighboring molecules beyond the limit distance, the structural integrity of the liquid will be violated, and cavities will be generated. Subsequently, a series of evolutionary action of cavitation bubbles appear under the ultrasonic field and the cavitation phenomenon is produced.

### 2.2. Ultrasonic Sound Pressure

During the UA-ELID grinding, the ultrasonic filed is induced under the high-frequency vibration of grinding wheel, and the ultrasonic waves propagate in the grinding medium. It is well known that the two pre-conditions for cavitation are (i) ultrasonic sound pressure surpassing the cavitation threshold and (ii) availability of cavitation bubble nuclei [23]. The first condition is ensured by the fact that the electrolyte permanently contacts the vibrating surface of the grinding wheel under the ultrasonic vibration, while the second one is provided by the presence of bubble nuclei and tiny droplets in the electrolyte. Once the ultrasonic sound pressure surpasses the cavitation threshold of electrolyte, the cavitation occurs during the UA-ELID grinding.

Considering facilitating the calculation, it was assumed that the cavitation bubbles satisfy the following conditions [24].

(1)Cavitation bubbles maintain a spherical shape throughout their motion.(2)Cavitation bubbles exist in an infinite space of liquid.(3)The temperature and density of the surrounding liquid are constant.(4)The effects of gravity and buoyancy on cavitation bubbles are neglected.(5)The effect of the chemical reaction on the internal bubble energy is not considered.(6)The effects of grinding force and grinding disturbance velocity on bubbles are taken into account.

In addition, the viscous effect of electrolyte is disregarded in calculations. According to the Bernoulli equation of potential flow, the relationship between the disturbance pressure p and the disturbance velocity v can be expressed as follows [25]:(1)$$p+\frac{1}{2}\rho_{e}\cdot v^{2}+\rho_{e}\cdot g\cdot h=c(t)$$
where $\rho_{e}$ is the density of the electrolyte.

Equation (1) can be rewritten as:(2)$$p=c(t)-\frac{1}{2}\rho_{e}v^{2}-\rho_{e}gh$$

In the previous study [11], the disturbance velocity of the grinding wheel was derived as follows:(3)$$v=v_{g}=\sqrt{{\left(v_{s}-v_{w}\right)}^{2}+v_{f}{}^{2}+{\left[f_{a}+A\cdot \sin(2\pi f+\varphi )\right]}^{2}}$$
where *A* denotes the ultrasonic amplitude, *f_a_* denotes the axial feed rate of grinding wheel, *f* denotes the ultrasonic frequency, and $v_{f}$ denotes the feed rate of workbench.

Substituting Equation (3) into Equation (2), the following equation can be obtained.
(4)$$p=c(t)-\frac{1}{2}\rho_{e}\cdot \left\{{\left(v_{s}-v_{w}\right)}^{2}+{v_{f}}^{2}+{\left[f_{a}+A\cdot \sin(2\pi ft+\varphi )\right]}^{2}\right\}-\rho_{e}\cdot g\cdot h$$

Therefore, the amplitude of pressure variation on the grinding wheel can be expressed as,
(5)$$\frac{\partial p}{\partial t}=-\rho_{e}\cdot A\cdot 2\pi f\cdot f_{a}\cdot \cos(2\pi ft+\varphi )-\frac{1}{2}\rho_{e}\cdot A^{2}\cdot 2\pi f\cdot \sin(4f\cdot t+2\varphi )$$

This implies that the density $\rho_{e}$ of the electrolyte, the axial feed rate $f_{a}$, the ultrasonic amplitude A and the frequency f determines the pressure p around the grinding wheel.

According to Liu [26], the disturbance pressure p satisfies the Laplace’s equation:(6)$$\nabla^{2}p=0$$

In this paper, the diameter of the grinding wheel was 25 mm, and the grain size is 3.5 to 40 μm. In addition, the ultrasonic frequency ranged from 20 to 35 kHz, and the ultrasonic amplitude was approximated at 10 μm, since the clearance between the grinding wheel and workpiece during processing was close to the prominent height of the grinding, and the effective grinding size was only 1/3 of the grinding particle size. Besides, considering wear of the grinding particle, the clearance between the wall and the grinding wheel was assumed to be 0.01 mm. According to Equation (6), the disturbance pressure was computed via the 2014 MATLAB software, and the computation results are shown in Figure 2.

The calculated disturbance pressure range was 4.58–5.27 MPa. According to the findings of Guo et al. [27], the cavitation threshold of most electrolyte was 1–3 MPa. As shown in Figure 2, the ultrasonic sound pressure is higher than the cavitation threshold of the electrolyte, which ensures that the ultrasonic cavitation is bound to occur on the grinding wheel surface. It is observed from Figure 2a that the fluctuation of pressure grows with the ultrasonic frequency. In Figure 2b, the ultrasonic sound pressure enhances with an increase of ultrasonic amplitude. This strongly indicates that the ultrasonic sound pressure is controlled by both parameters, but the effect of ultrasonic amplitude is more pronounced than that of ultrasonic frequency. Therefore, the influence of ultrasonic amplitude on the cavitation effect will be explored in the following section.

### 2.3. Collapse of Cavitation Bubbles

During the UA-ELID grinding, the bubbles generating with the help of cavitation effect would be continuously compressed and stretched by the electrolyte under the high-frequency vibration. When the ultrasonic pressure amplitude exceeds the cavitation threshold, the bubbles will eventually collapse. At this time, the electrolyte around each bubble will rush to its center at a higher velocity due to the reactive force and exert greater impact on the grinding wheel. Therefore, it is essential to analyze the collapse of bubbles during the UA-ELID grinding.

It is supposed that the cavitation bubbles are collapsed in the compression model. Based on the law of energy conservation, the liquid velocity with the cavitation bubble closure can be expressed as follows [25].
(7)$$4\pi r^{2}v(r,t)=4\pi R\prime R^{2}$$
where *R* is the bubble radius, *R′* is the liquid radial speed at the bubble boundary.

In order to facilitate the calculation, the viscous effect of liquid is ignored, and the motion of liquid around the bubbles can be regarded as a potential flow. At the infinite distance, the potential function *K* can be obtained using the following formula.
(8)$$K=\int_{r}^{\infty }vdr=-\frac{R\prime R^{2}}{r}$$

During the UA-ELID grinding, the surrounding pressure acting on the bubble *p_∞_* can be obtained using the following formula.
(9)$$p_{\infty }=p_{0}+p_{g}+p_{A}\sin2\pi ft$$
where $p_{0}$ denotes the standard atmospheric pressure, $p_{g}$ and $p_{A}$ denotes grinding pressure and ultrasound pressure, respectively.

According to Plesset et al. [28], the liquid pressure can be obtained as follows.
(10)$$p-p_{\infty }=-\rho_{e}(\frac{\partial K}{\partial t}+\frac{1}{2}v_{r}{}^{2})$$

Substituting Equation (8) into Equation (10), the latter can be expressed as follows.
(11)$$p-p_{\infty }=2\rho_{e}{\left(\frac{R}{r^{2}}\right)}^{2}\left(\frac{r^{3}}{R}{R\prime }^{2}+\frac{1}{2}r^{3}R\prime \prime -\frac{1}{4}R^{2}R\prime^{2}\right)$$

When the disturbance pressure p is equal to the vaporized liquid pressure $p_{v}$ at the boundary of bubble; namely, *r* = *R*, Equation (11) can be rewritten as follows.
(12)$$p_{v}-p_{\infty }=2\rho_{e}{\left(\frac{1}{R}\right)}^{2}\left(\frac{3}{4}R^{2}{R\prime }^{2}+\frac{1}{2}R^{3}R\prime \prime \right)$$

It is assumed that the $p_{v}-p_{\infty }$ difference remains unchanged during the bubble closure with disregard of the tension and viscous flow effect, and thus the Equation (12) can be integrated.
(13)$$\left(p_{v}-p_{\infty }\right)\frac{dR^{3}}{dt}=\frac{3}{2}\rho_{e}\frac{d(R^{3}{R\prime }^{2})}{dt}$$

The time of bubble closure t can be derived using the following formula.
(14)$$t=\int_{R}^{R_{0}}\frac{1}{R\prime }dt=\sqrt{\frac{3\rho_{e}}{2R}}\int_{R}^{R_{0}}\frac{R^{2}}{\sqrt{\left(p_{v}-p_{\infty }\right)\left(R_{0}^{3}-R^{3}\right)}}dt$$

Taking $\lambda =R/R_{0}$, the Equation (14) can be reduced to the following form.
(15)$$t=\sqrt{\frac{3{R_{0}}^{2}\rho_{e}}{2\left(p_{v}-p_{\infty }\right)}}\cdot \int_{\lambda }^{1}\sqrt{\frac{\lambda^{3}}{1-\lambda^{3}}}d\lambda$$

The further formulas use the gamma-function and the following equation:(16)$$\int_{0}^{1}x^{\alpha }{\left(1-x\right)}^{\beta }dx=\frac{\Gamma (\alpha +1)\cdot \Gamma (\beta +1)}{\Gamma (\alpha +\beta +2)}$$
where Γ denotes the gamma Function.

At R=0 and λ=0, the bubble would absolutely collapse, and the time t′ of the bubble completely collapsed can be obtained as follows.
(17)$$t\prime =\sqrt{\frac{3{R_{0}}^{2}\rho_{e}}{2\left(p_{v}-p_{\infty }\right)}}\frac{\Gamma (3/2)\Gamma (1/2)}{\Gamma (2)}\approx 1.923\sqrt{\frac{{R_{0}}^{2}\rho_{e}}{p_{v}-p_{\infty }}}$$

According to Guo et al. [29], under the laboratory conditions, the bubble nuclei size was approximately 5–20 μm, and thus in this study the size of cavitation bubble generated on the surface of grinding wheel was in this range. During the processing, because the electrolyte is continuously poured on the grinding wheel, it is assumed that the surface of the grinding wheel should be immersed in the still water, and the $p_{v}$ is 2.3 times larger than the atmosphere pressure. Substituting the above parameters and the density of water of 1 × 10^3^ kg/m^3^ into Equation (17), the time of the cavitation bubble collapse in the UA-ELID grinding can be calculated to be about 1.26–5.31 μs. It occupies 2.5–20% of the period of ultrasonic vibration, with the frequency range from 20 to 35 kHz. Therefore, the collapse of cavitation bubbles will occur several (5 to 40) times per each vibration cycle.

Combining Equations (14) and (17), the time *t’* can be calculated as follows.
(18)$$\frac{t}{t\prime }\approx 0.637\int_{\lambda }^{1}{\left(\lambda^{3}\right)}^{1/2}{\left(1-\lambda^{3}\right)}^{1/2}d\varepsilon$$

The relationship between t/t′ and λ is computed via the MATLAB software, and the results are shown in Figure 3a. It can be observed from Figure 3a that the time of cavitation bubble collapsed enhances with an increase of the initial radius. According to Equation (13), the velocity of liquid on the bubble surface along the radical direction is depicted in Figure 3b: when cavitation bubbles are fully collapsed, the liquid velocity tends to infinity. It is attributed to the fact that the liquid is assumed to be incompressible and by the energy conservation law, the velocity of bubbles must tend to infinity. Moreover, the pressure in the bubble is supposed to be constant, and thus there will be no obstacles to the grinding liquid to flow during the bubble collapse. At λ=0.1 and $p_{v}-p_{\infty }=0.13\;\mathrm{MPa}$, the maximum pressure with the cavitation bubble collapse approximately reaches 43 MPa. According to Equation (7), the radial velocity at the bubble boundary is about 245 m/s. It can be seen that the ultrasonic vibration enhances the instantaneous impact of the cavitation bubble collapse, including a high-speed micro-jet, and exhibits a significant effect on the grinding wheel and processed workpiece surface.

### 2.4. Cavitation Bubble Radius

Considering the disturbance pressure and grinding velocity effects on the cavitation bubble radius *R*, the latter parameter is substituted into Equation (13), the original equation is expanded to Equation [25]:(19)$$r\prime \prime +r\frac{1}{\rho_{e}\cdot {R_{0}}^{2}}\left[3k\left(p_{0}+\frac{2\sigma }{R_{0}}\right)-\frac{2\sigma }{R_{0}}\right]=\frac{p_{A}\cdot \sin(\omega \cdot t)-p_{g}}{\rho_{e}\cdot R_{0}}-\frac{3{v_{g}}^{2}}{2R_{0}}$$

If the vibration of cavitation bubble is approximated by the spring subsystem vibration without damping, according to the vibration mechanics, the following equation can be acquired.
(20)$$\begin{array}{cc} r\prime \prime +r\cdot {\omega_{d}}^{2} & =\frac{p_{A}\sin(\omega t)-p_{g}}{\rho_{e}R_{0}}-\frac{3{v_{g}}^{2}}{2R_{0}}\\ & =\frac{p_{A}\sin(\omega t)}{\rho_{e}R_{0}}-\frac{p_{g}}{\rho_{e}R_{0}}-\frac{3}{2R_{0}}\left[{\left(v_{s}-v_{w}\right)}^{2}+{v_{f}}^{2}+{f_{a}}^{2}\right]-\frac{3}{R_{0}}A\cdot \omega \cdot f_{a}\cdot \cos(\omega t+\varphi )\\ & -\frac{3}{2R_{0}}A^{2}\omega^{2}\cos^{2}(\omega t+\varphi ) \end{array}$$

Under the initial condition $r_{i}(0)=0$, Equation (20) is equivalent to the superposition of the following four equations.
(21)$$\left\{ \begin{array}{l} r_{1}\prime \prime +r_{1}\cdot {\omega_{d}}^{2}=\frac{p_{A}\sin\omega t}{\rho_{e}R_{0}}\\ r_{2}\prime \prime +r_{2}\cdot {\omega_{d}}^{2}=-\frac{p_{g}}{\rho_{e}R_{0}}-\frac{3}{2R_{0}}\left[{\left(v_{s}-v_{w}\right)}^{2}+{v_{f}}^{2}+{f_{a}}^{2}\right]\\ r_{3}\prime \prime +r_{3}\cdot {\omega_{d}}^{2}=-\frac{6}{R_{0}}A\cdot \pi f\cdot f_{a}\cdot \cos(\omega t+\varphi )\\ r_{4}\prime \prime +r_{4}\cdot {\omega_{d}}^{2}=-\frac{3}{2R_{0}}A^{2}\cdot \omega^{2}\cdot \cos^{2}(\omega t+\varphi ) \end{array}\right.$$

In the above four equations, the first one describes the dynamic variation of cavitation caused by the propagation of ultrasound in the electrolyte, while the second one defines the cavitation caused by multiple factors, such as grinding pressure and speed under specific processing. The third equation covers the dynamic fluctuation of the bubble caused by the axial reciprocation of the grinding wheel and different phases of the ultrasonic wave. The fourth equation refers to the dynamic fluctuation of the bubble caused by ultrasonic waves at different phases and resonant frequency. They can be solved analytically. The solution of the above four nonlinear vibration equations can be composed of the general solution of the corresponding homogeneous equation and particular solution of a non-homogeneous equation, as follows.
(22)$$\left\{ \begin{array}{l} r_{1}=\frac{p_{A}}{\rho_{e}\cdot R_{0}\cdot (\omega_{d}-\omega )}\left[\sin(\omega \cdot t)-\frac{\omega }{\omega_{r}}\sin(\omega_{r}\cdot t)\right]\\ r_{2}=-\frac{\cos(\omega_{d}\cdot t-1)}{{\omega_{d}}^{2}}\left\{\frac{p_{g}}{\rho_{e}\cdot R_{0}}+\frac{3}{2R_{0}}\left[{\left(v_{s}-v_{w}\right)}^{2}+{v_{f}}^{2}+{f_{a}}^{2}\right]\right\}\\ r_{3}=-\frac{3A\cdot \omega \cdot f_{a}}{2R_{0}\cdot ({\omega_{d}}^{2}-\omega^{2})}[\cos(\omega_{d}\cdot t+\varphi )-\cos(\omega \cdot t+\varphi )]\\ r_{4}=-\frac{3A^{2}\cdot \omega^{2}}{2R_{0}\cdot ({\omega_{d}}^{2}-\omega^{2})}[\cos^{2}(\omega_{d}\cdot t+\varphi )-\cos^{2}(\omega \cdot t+\varphi )] \end{array}\right.$$

Based on the superposition principle, namely $r=r_{1}+r_{2}+r_{3}+r_{4}$, the radius of the cavitation bubble can be presented as *R* = *R* + *r*_0_. According to Lopita’s law, as $\omega \to \omega_{d}$ and φ=0, the solution of the equation under the cavitation bubble resonance can be obtained as follows.
(23)$$\left\{ \begin{array}{l} r_{1}=\frac{p_{A}}{2\rho_{e}R_{0}\omega_{d}}\left[\sin(\omega_{d}t)-\omega_{r}t\cos(\omega_{d}t)\right]\\ r_{2}=-\frac{\cos(\omega_{d}t)-1}{{\omega_{d}}^{2}}\left\{\frac{p_{g}}{\rho_{e}R_{0}}+\frac{3}{2R_{0}}\left[{\left(v_{s}-v_{w}\right)}^{2}+{v_{f}}^{2}+{f_{a}}^{2}\right]\right\}\\ r_{3}=-\frac{3Af_{a}t}{4R_{0}}\sin(\omega_{d}t)\\ r_{4}=-\frac{3A^{2}\omega_{d}t\sin(2\omega_{d}t)}{2R_{0}} \end{array}\right.$$

Equation (23) is solved numerically. Assuming $R=R_{0}+r_{i}$(*i* = 1, 2, 3, 4), four variation patterns of the single cavitation bubble radius in the grinding zone are obtained and plotted in Figure 4. At *i* = 1, the cavitation bubble performs a periodic motion near the equilibrium position and is close to steady state. At *i* = 2, its amplitude is lower than that in the other three groups, and the vibration period is also the shortest. In cases of *i* = 3 and *i* = 4, the variation amplitude is basically consistent, while the vibration period at *i* = 3 is lower than that at *i* = 4. Therefore, the peculiar processing environment to the grinding zone plays an important part in the generation and collapse of the cavitation bubble.

Synthesizing various influencing factors in Figure 4, the dynamic variation of a single cavitation bubble in the grinding zone is presented in Figure 5. It can be seen that the cavitation bubble undergoes a dynamic process in the grinding zone, including generation, collapse, regrowth and re-collapse in a short time. Because the dynamic model of the single cavitation bubble does not consider the viscous damping and the time lag of the liquid medium, the dynamic law of the single cavitation bubble evolution in the Figure 5 shows a process similar to the steady state cavitation. However, as the processing time continues, the cavitation bubble radius experiences cyclic compressing and stretching, and the overall radius decreases.

## 3. Experiments

### 3.1. Experimental Setup and Conditions

The UA-ELID grinding tests were carried out on a machining center (VMC850E) through superimposing the ultrasonic vibration system. Figure 6 presents the experimental setup, which leadingly contains the ultrasonic vibration system, the ELID power supply, a process parameter monitoring system, the EDM (electron discharge machining) dressing device, and the ELID electrolysis device. The vibration system was immobilized on the machine tool spindle with a clamp. It was composed of the ultrasonic generator, electric wireless transmission device, transducer, horn, and diamond grinding wheel. The ELID power supply mainly provided the energy for the processing and controlled some parameters. The process parameter monitoring system was utilized to observe the stability of the processing. The EDM dressing device fixed on the left side of the workbench was utilized to pre-dressing the diamond grinding wheel. The ELID electrolysis device was fixed on the spindle through the special fixture to ensure the online dressing the grinding wheel. The three-dimensional (3D) microscopy system (VW-6000, KEYENCE, Osaka, Japan) was utilized for online monitoring of the ultrasonic cavitation effect. During the observation process, a 3D microscope system and a high-speed CMOS camera was used to observe and record the cavitation process, respectively. This online system could realize online observation, slow down and magnify the monitored process, and capture the generation, movement, and collapse of bubbles on the surface of the grinding wheel.

The workpiece was fabricated from the ZTA ceramics with 20% zirconia, and its highest sintering temperature reached 1450 °C. The relevant parameters are presented in Table 1. During the experiment, the ultrasonic amplitude was changed by adjusting the ultrasonic power. In the experiment, the C-ELID and UA-ELID grinding were achieved by switching the ultrasonic generator on or off.

The grinding wheel surface was observed using the ultra-depth 3D microscope (VHX-2000, KEYENCE, Osaka, Japan) to obtain the variation of grain distribution. The 3D white light interference (Talysurf CCI6000, Taylor Hobson, Leicester, UK) was employed to obtain the number of grains in the unit area, observe the morphology of the grinding wheel and measure the surface roughness of the workpiece. An index called the surface vertex arithmetic mean curvature ($S_{SC}$) was introduced to quantitatively reflect the grain sharpness and evaluate the dressing effect of grinding wheel surface. It can be expressed as follows.
(24)$$S_{SC}=-\frac{1}{2n}\sum_{k=1}^{n}\left(\frac{\partial z^{2}(x,y)}{\partial x^{2}}+\frac{\partial z^{2}(x,y)}{\partial y^{2}}\right)$$

The surface residual stress was measured via the XRD method using a PROTO X-ray device. The surface morphology was observed via the scanning probe microscope (CSPM-2000, Being Nano-Instruments Co., Ltd., Guangzhou, China). The X-ray diffractometer (D8ADVANCE, Bruker, Billerica, MA, USA) was utilized to analyze the surface of the machined workpiece. In order to gain reliable data, three points were gauged for each set of parameters, and their mean values were used as the final results.

### 3.2. On-Line Monitoring Cavitation Effect

In order to analyse the cavitation effect on the grinding wheel, selecting any video recorded by the high-speed CMOS camera during the UA-ELID grinding. Figure 7 illustrates the online observation results of the cavitation effect. It was found that the cone-like bubble structure (CBS) appeared on the grinding wheel surface. Cavities in the sound field were mainly affected by two forces; namely, the main Bjerknes force, and the second-order Bjerknes force. The former is the direct-acting force of ultrasonic waves on cavities, and the latter is the interaction force between individual cavities [30]. In general, the main Bjerknes force is much larger than the second-order Bjerknes force. Ye et al. [31] considered that the formation of CBS structure was related to the main Bjerknes force. When the sound pressure amplitude increases, the main Bjerknes force changes direction, causing the high-pressure region that attracts the cavities to become a repulsion region. Due to the repulsion effect of the main Bjerknes force, the cavities cannot enter this area but pass around the area where the main Bjerknes force does not exist. In addition, the CBS moved along the grinding wheel surface with the increasing of time.

During the monitoring process, as shown in Figure 8, the variation of the CBS on the grinding wheel surface was observed under different amplitudes. It was found that both the number of CBSs on the grinding wheel surface and the radius of the cavitation bubbles increased with the ultrasonic amplitude. This can be attributed to the fact that the ultrasonic energy was positively related to the amplitude as the ultrasonic frequency kept constant. With the ultrasonic amplitude increasing, the more ultrasonic energy was absorbed by the electrolyte, and the easier ultrasonic cavitation took place on the grinding wheel surface. This was a good agreement with the results presented in the Section 2.2. In addition, the cavitation bubbles absorbed more energy from the negative ultrasonic pressure, and led to its radius increase, namely the λ enhanced. According to Figure 4a, the larger λ values indicated that the time period of cavitation closure was greatly shortened. Therefore, a stronger impact from the cavitation collapse would be observed on the grinding wheel surface at the larger amplitude.

### 3.3. Effect of Ultrasonic Cavitation on the Oxide Film

During the UA-ELID grinding, since the grinding wheel was dressed online, an oxide layer namely oxide film could be formed on the grinding wheel. As shown in Figure 9, with the help of ultrasonic vibration, the cavitation effect would erect a certain effect on the film. It was found that the micro pits appeared on the grinding wheel surface. They were not only generated in the thickest oxide film, but also in a thinner area, and the size of the micro pits was different. This proved that the cavitation bubbles grew on the grinding wheel surface until their collapse, which made a dramatic impact on the grinding wheel and workpiece surfaces. The ultrasonic cavitation phenomenon permanently manifested itself during the UA-ELID grinding, which also proved the correctness of the previous theoretical analysis. Moreover, the influence of ultrasonic amplitude on the oxide film was presented in Figure 9. As the ultrasonic amplitude increasing, the oxide film became gradually thinner, but remained relatively dense. This was mainly attributed to the fact that the ultrasonic cavitation was enhanced with the ultrasonic amplitude, and the vibration of the grinding wheel was improved as well. The coupled effects improved the oxide film’s relative dense and the adhesion.

## 4. Results and Discussions

### 4.1. Dressing Effect of the Grinding Wheel

#### 4.1.1. Number and Distribution of Diamond Grains

Figure 10 presented the number of grains in a unit area of 1 mm^2^ and *Ssc* under different ultrasonic amplitude, in which the ultrasonic amplitude 0 denoted the C-ELID grinding. As seen in Figure 10, the number of grains during the UA-ELID grinding was larger than that of the C-ELID grinding, and grain sharpness was improved to a certain extent. Simultaneously, with the increase of the ultrasonic amplitude, both the number of grains and their sharpness were elevated. The number of grains at higher ultrasonic amplitude was approximately twice higher than that of C-ELID grinding, and the sharpness increased by 43.2%. This explains why high-frequency ultrasonic vibration easily caused the cavitation on the grinding wheel surface and changed the mechanism of oxide film generation. Simultaneously, under the ultrasonic vibration, the higher impact from the cavitation bubble collapse would act on the grinding wheel surface, and thus the oxide film thickness declined, as compared to that of the C-ELID grinding and reduce the number of diamond grains covered by the oxide film. In addition, the cavitation effect became stronger at larger ultrasonic amplitude, increasing the instantaneous impact force. This, in turn, would enhance the effective cutting edges of diamond grains and meanwhile increased their sharpness.

#### 4.1.2. Surface Morphology of the Grinding Wheel

Figure 11 presents the 3D morphology of the grinding wheel surface under C-ELID and UA-ELID processing conditions. It can be seen that the grain protrusion height in the former case was higher than that in the latter one. During the C-ELID grinding, the diamond grains was buried in the oxide film, so the number of grains protruding from the grinding wheel surface was lower than in UA-ELID, but the maximum protrusion height of grains was larger. Under the UA-ELID grinding, the higher vibration can lead to the oxide film detachment from the grinding wheel surface, and provide its dressing, which would decrease the maximum protrusion height and grain spacing [25]. Besides the online dressing, the UA-ELID grinding combines the high-frequency ultrasonic vibration with the cavitation effect, which jointly enhanced the oxide film removal. Therefore, the oxide film on the grinding wheel surface was relatively thinner, and the relative number of grains was larger in UA-ELID than in C-ELID grinding.

#### 4.1.3. Grinding Wheel Surface

The dressing effect of the grinding wheel during both conditions is presented in Figure 12. It was observed from Figure 12 that the average protrusion height of grains during the UA-ELID grinding was exceeded that in the C-ELID grinding by 14.2%. However, the average grain spacing in the former case was lower than that in the latter by 21.2%. This can be attributed to the fact that the ultrasonic vibration could decline the oxide film thickness on the grinding wheel, which was conducive to enhance the average protruding height of grains. In addition, the ultrasonic vibration could improve the maximum grinding force, and it was beneficial to damage the grains and forming new cutting edges. The joint action of these factors implies the above effect on the average protrusion height of grains. Insofar as the number of grains per unit area in the UA-ELID grinding was larger than that in the C-ELID, the average grain spacing in the former case was lower, respectively. As the ultrasonic amplitude increased, the ultrasonic cavitation was intensified, as well as its action on grains, which exerted a significant effect on the grinding wheel surface.

### 4.2. Grinding Effect of Workpiece

#### 4.2.1. Surface Roughness of the Workpiece

The surface roughness of the workpiece under both processing was shown in Figure 13. It can be seen from the Figure 13 that the surface roughness decreased with the ultrasonic amplitude increasing. Compared with the C-ELID grinding, the surface roughness parameters of *R*z and *R*a maximum dropped by 54.2% and 46.5%, respectively, during the UA-ELID grinding, which was the reason that the increase of ultrasonic amplitude strengthened the ultrasonic cavitation, and its action on the grinding wheel surface became more uniform. When the ultrasonic amplitude increased, the micro-jet induced by the rapid generation and collapse of cavitation bubbles would exhibit a scouring effect on the grinding wheel surface. At the same time, the motion path of diamond grains was also changed, and the average grinding force was reduced at higher amplitude. This would result in surface roughness decline and surface quality improvement.

#### 4.2.2. Surface Morphology of Workpiece

The surface morphology under different ultrasonic amplitude was presented in the Figure 14. The grooves on the surface processed by C-ELID grinding were more pronounced, while the surface was getting gradually smoother with the ultrasonic amplitude increasing. It was demonstrated that the materials removal occurred by plastic deformation under the ultrasonic vibration. The XRD scanning was utilized to observe the workpiece surface under different ultrasonic amplitude. The results of different ultrasonic amplitude were respectively presented in Figure 15. As presented in Figure 15, the ZTA ceramics underwent the martensite transformation, in which *t*-ZrO2 particles transformed to *m*-ZrO2 particles with the help of ultrasonic vibration. In addition, 3.6% of *t*-ZrO2 particles took part in the transformation, which led to the volume expansion by 3–5%. In other words, during the UA-ELID grinding, the material removal was changed to a certain extent. According to Tong et al. [32], the characteristics of nanocomposite ceramic were changed by the vibration induced stress. During the ultrasonic vibration, the high internal vibration-induced stress significantly reduced the equivalent hardness and softened the surface layer. As shown in Figure 15, the number of micro pits on the workpiece surface gradually augmented with an increase in ultrasonic amplitude. These results indicated that the ultrasonic cavitation induced a high instantaneous impact force on the surface. The higher ultrasonic amplitude was applied to the surface, the more uniform is the distribution of micro pits on the surface. This also manifested that the ultrasonic cavitation erected a significant effect on the mechanism of material removal.

#### 4.2.3. Surface Residual Stress of the Workpiece

Figure 16 presented the variation of the surface residual stress of nanocomposite ceramic under C-ELID and UA-ELID grinding. As shown in Figure 16, the residual compressive stress was introduced during the UA-ELID grinding and C-ELID grinding. Compared with the latter, the residual compressive stress maximum increased by 44.5% during the former. It is well known that the surface residual stress was generated by the coupling effect of the material phase transformation, thermal plastic deformation produced by the grinding heat and cold plastic deformation by the mechanical stress. The ZTA ceramics belongs to the nanocomposite ceramics with phase transition. During the processing, the higher grinding temperature and grinding force would cause the phase transition stress, and lead to the volume expanding by 3–5%, which could introduce the high residual compressive stress into the surface. In other words, the higher martensite phase transformation could produce the greater residual compressive stress. As shown in Figure 15, the higher the t-m phase transformation rate ZTA nanocomposite ceramics was taken place during the UA-ELID grinding. As a result, relatively high residual compressive stress could be generated in the transformation layer. In addition, the higher ultrasonic amplitude with stronger cavitation would act on the surface of the workpiece, and the rapid collapse of cavitation bubbles would have a stronger impact on the workpiece surface. The instantaneous impact could promote the martensite phase transformation and generate high residual stress. Simultaneously, the ultrasonic vibration also changed the diamond grain motion path and caused the flank face of grains reciprocating ironing on the workpiece surface. Therefore, the residual compressive stress could be improved to a certain extent.

## 5. Conclusions

In this paper, the cavitation effect in the UA-ELID grinding was investigated in order to explore its action on the diamond grinding wheel and nanocomposite ceramic workpiece surface. The theoretical models of the cavitation threshold, collapse and radius variation of cavitation bubbles were elaborated and experimentally verified. During the actual grinding test, the surface characteristics for the diamond grinding wheel and nanocomposite ceramic workpiece during the UA-ELID grinding were compared with those of the C-ELID. According to the findings, the following conclusions can be drawn.

(1)It was found that the disturbance pressure was higher than the cavitation threshold under the ultrasonic vibration action. The duration of cavitation bubble closure decreased with an increase in its initial radius, and after the complete closure, the velocity of liquid in bubbles would be toward to infinity. Considering the influence of grinding force and grinding disturbance velocity on bubbles, as the processing time continues, the radius of the cavitation bubble changes during cyclical compression and tension, with the eventual decline of its overall value.(2)It was observed that, during the UA-ELID grinding, the ultrasonic cavitation easily occurred on the grinding wheel surface with the increase of ultrasonic amplitude. Meanwhile, the stronger instantaneous impact force induced by the cavitation bubble collapse would act on the processed surface at the larger ultrasonic amplitude. Combination with the grinding wheel vibration, the cavitation effect could improve the dense and adhesion of oxide film on the grinding wheel surface.(3)The dressing effect of the grinding wheel was found to be enhanced by the ultrasonic cavitation. With the increase of ultrasonic amplitude, the micro-jet derived from the cavitation bubbles rapid generation and collapse would have a scouring effect on the surface of the grinding wheel. It led to a more homogenous distribution of diamond grains protruding from the grinding wheel surface. Moreover, the number of grains per unit in the UA-ELID grinding increased approximate twice times than that in the C-ELID grinding, and the sharpness of grain was enhanced by 43.2%. The average protrusion height increased by 14.2%, while the average grain spacing decreased by 21.2%.(4)(4) As compared to the C-ELID grinding, the surface roughness of workpiece *R_z_* and *R_a_* decreased by 54.2% and 46.5% respectively during the UA-ELID grinding. Under the cavitation effect, the mechanism of material removal was changed, and the martensite transformation occurred during processing. Therefore, the material removal by the plastic deformation was enhanced, and the surface quality was improved. Simultaneously, the surface residual compressive stress increased by 44.5%.

## Figures and Tables

**Figure 1 materials-14-05611-f001:**
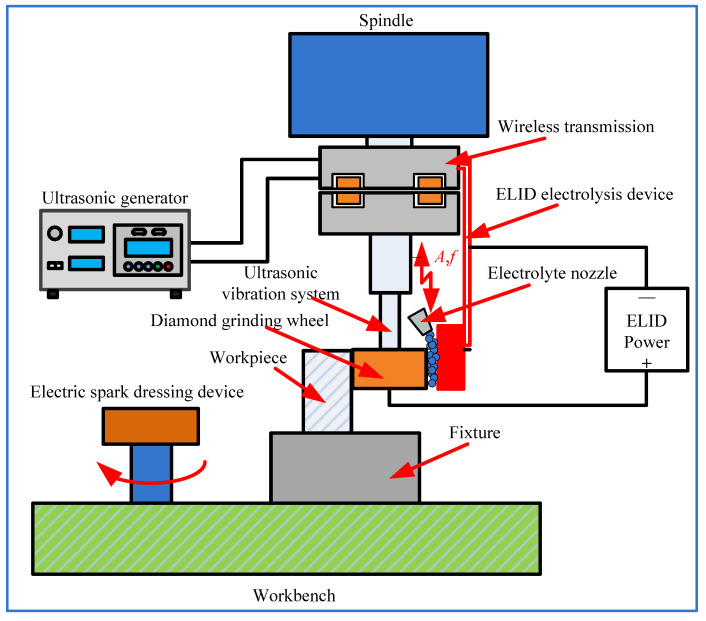
Schematic of UA-ELID grinding.

**Figure 2 materials-14-05611-f002:**
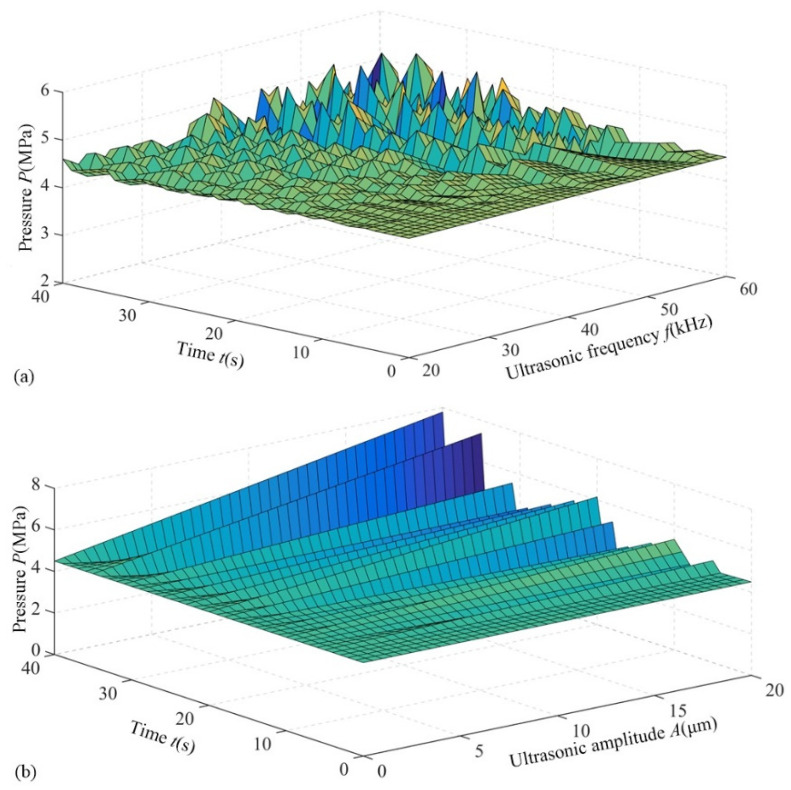
Variation of ultrasonic sound pressure. (**a**) Ultrasonic frequency; (**b**) Ultrasonic amplitude.

**Figure 3 materials-14-05611-f003:**
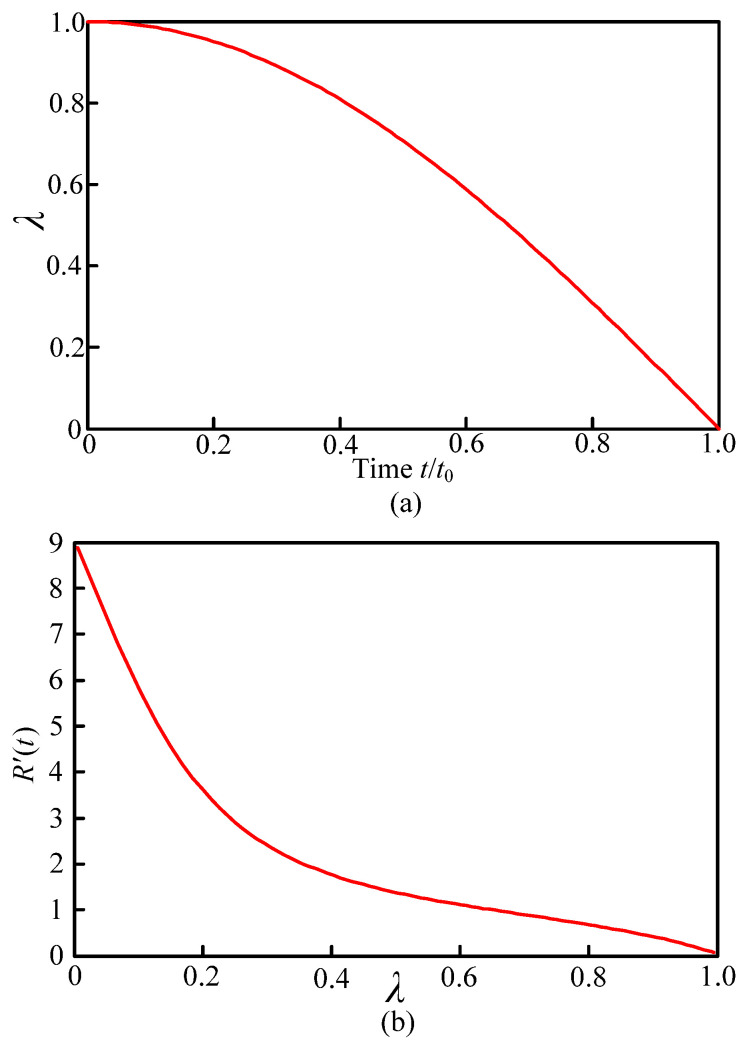
Evolution of λ (**a**) and *R′(t)* (**b**) with time.

**Figure 4 materials-14-05611-f004:**
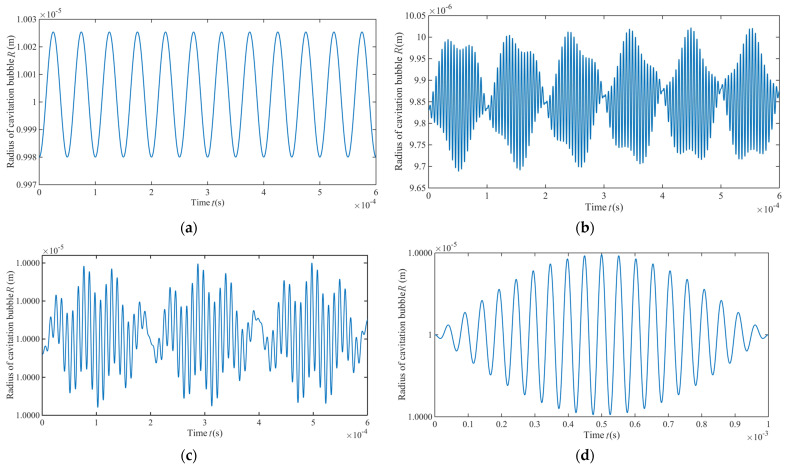
Variation trends of a single cavity radius in the resonance grinding zone. (**a**) *i* = 1; (**b**) *i* = 2; (**c**) *i* = 3; (**d**) *i* = 4.

**Figure 5 materials-14-05611-f005:**
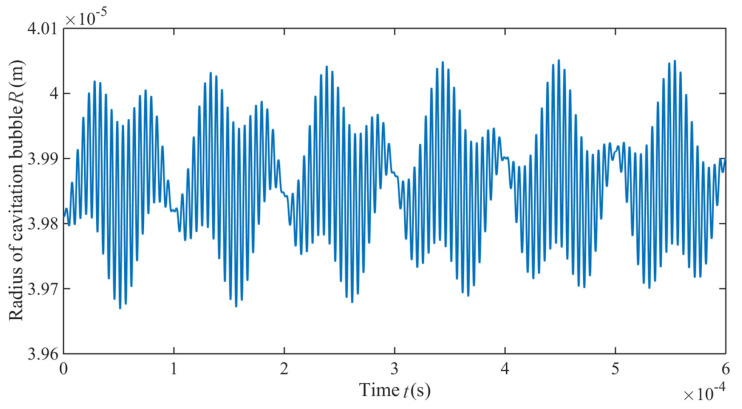
Overall trend of the single cavity radius in the grinding zone.

**Figure 6 materials-14-05611-f006:**
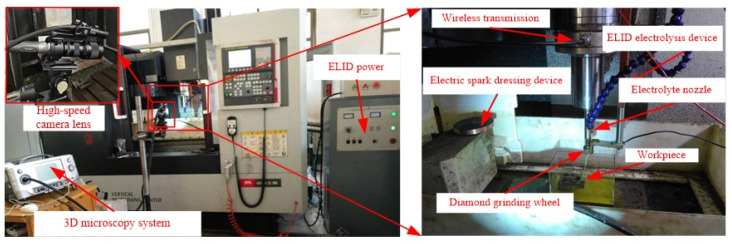
Experimental setup for UA-ELID grinding.

**Figure 7 materials-14-05611-f007:**
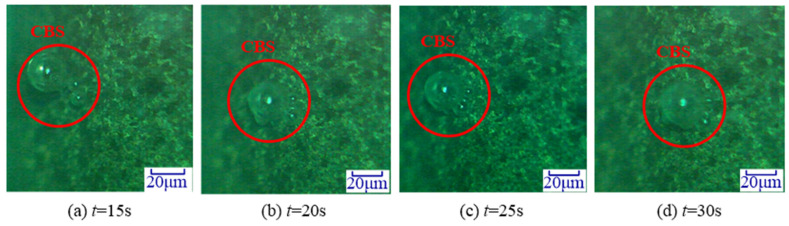
Cavitation on grinding wheel surface.

**Figure 8 materials-14-05611-f008:**
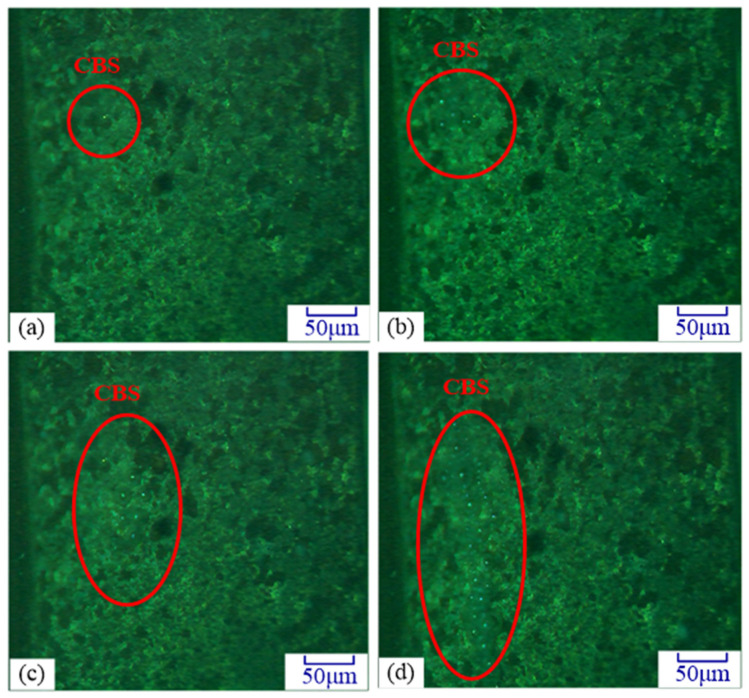
Effect of ultrasonic amplitude on the cavitation. (**a**) *A* = 6.4 μm; (**b**) *A* = 8.3 μm; (**c**) *A* = 10.2 μm; (**d**) *A* = 12.5 μm.

**Figure 9 materials-14-05611-f009:**
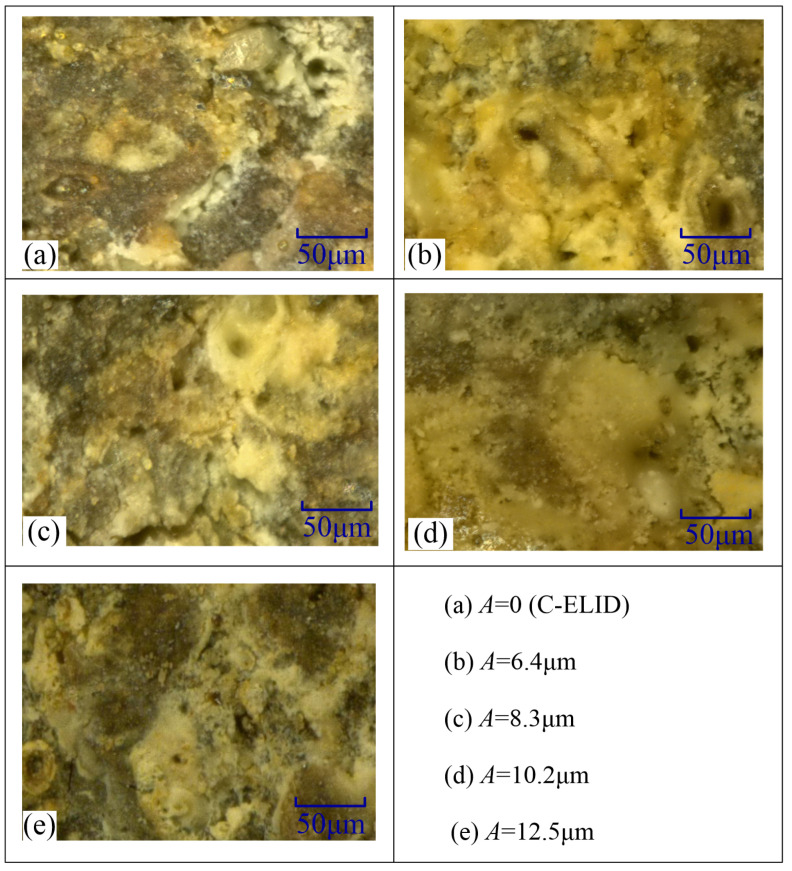
Effect of ultrasonic cavitation on the grinding wheel surface.

**Figure 10 materials-14-05611-f010:**
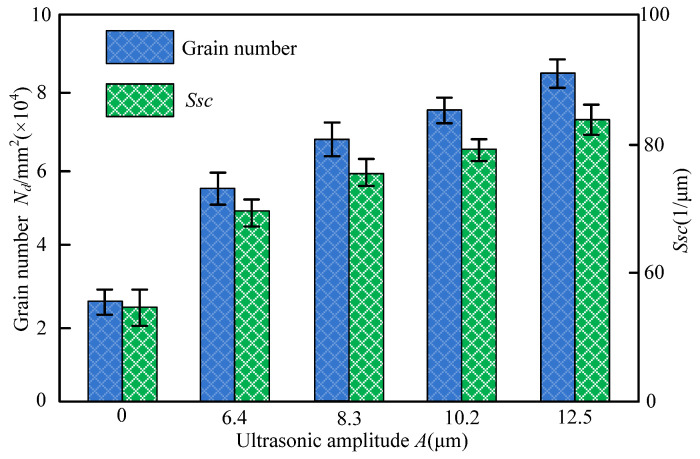
Variation of the number of grains and *Ssc* with ultrasonic amplitude.

**Figure 11 materials-14-05611-f011:**
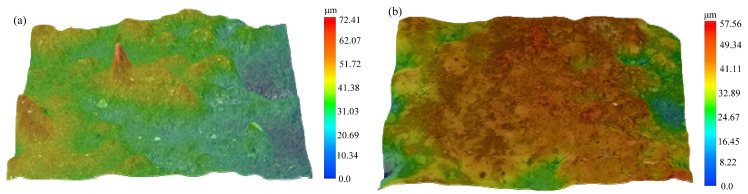
3D morphology of grinding wheel surface. (**a**) *A* = 0 μm, C-ELID; (**b**) *A* = 12.5 μm, UA-ELID.

**Figure 12 materials-14-05611-f012:**
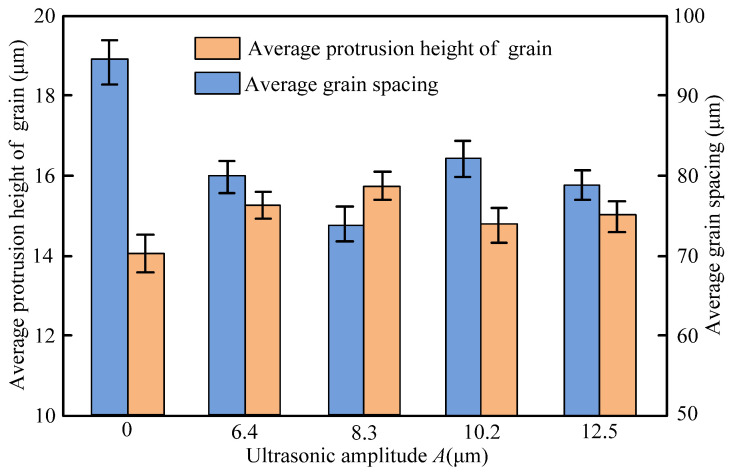
Average protrusion height and spacing of grains on the wheel surface under different ultrasonic amplitude.

**Figure 13 materials-14-05611-f013:**
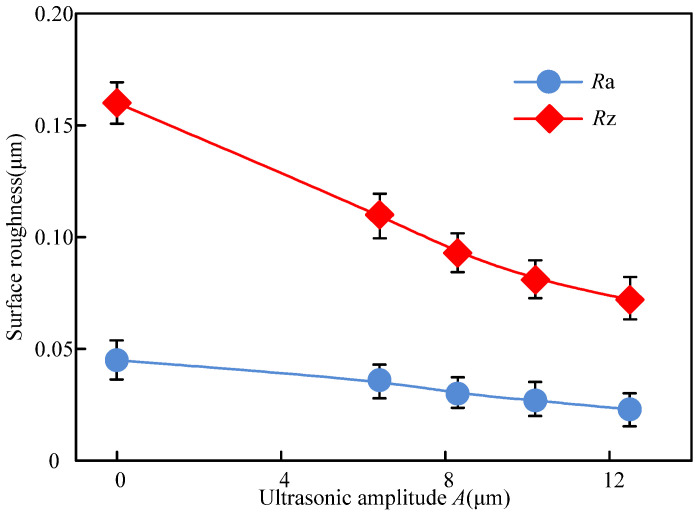
Variation of surface roughness with ultrasonic amplitude.

**Figure 14 materials-14-05611-f014:**
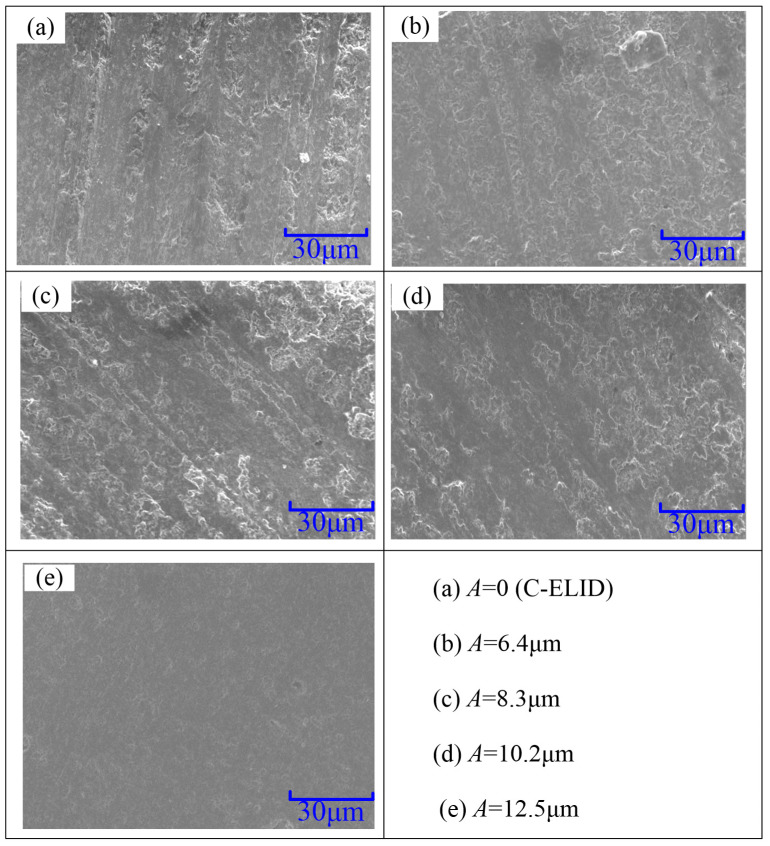
Surface morphology of the processed workpiece.

**Figure 15 materials-14-05611-f015:**
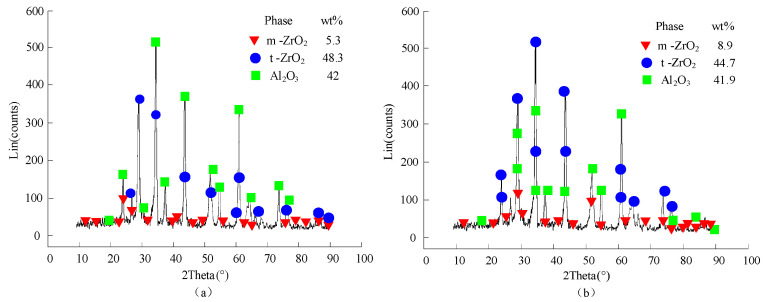
Results of XRD diffraction. (**a**) *A* = 0 μm (**b**) *A* = 12.5 μm.

**Figure 16 materials-14-05611-f016:**
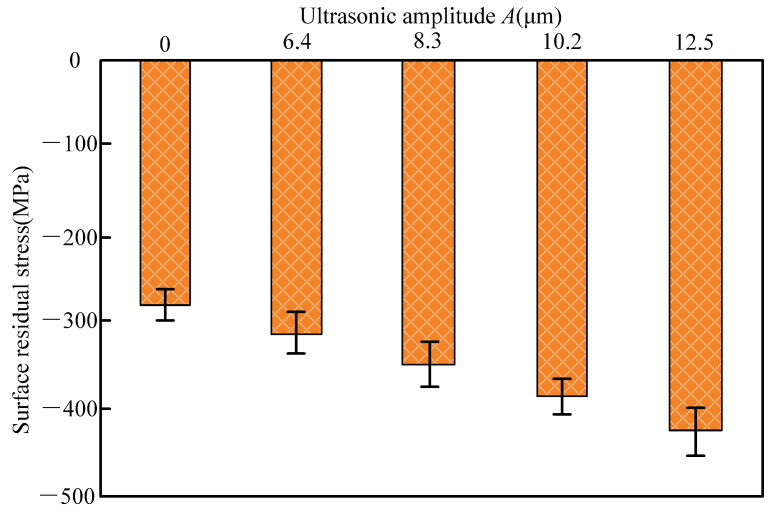
Variation of surface residual stress with ultrasonic amplitude.

**Table 1 materials-14-05611-t001:** Relevant parameters during grinding.

Types	Parameters	Value
Grinding wheel	Material	Cast iron bond diamond wheel
	Concentration	100%
	Model	W40
	Outer diameter (mm)	25
	Width (mm)	17
Grinding liquid	The ratio of mother liquid to distilled water	1:50
Triming	Voltage (V)	120
	Rotating speed (r/min)	1000
Workpiece	Length × width × height (mm^3^)	16 × 16 × 8
	Wheel speed (m/s)	2.6
Grinding parameters	Grinding depth (μm)	3
	Feed rate (mm/min)	100
Ultrasonic parameters	Frequency (kHz)	25.3
	Amplitude (μm)	0, 6.4, 8.3, 10.2, 12.5

## Data Availability

Not applicable.

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
