# Peer review of "Cavitation Effect in Ultrasonic-Assisted Electrolytic In-Process Dressing Grinding of Nanocomposite Ceramics"

_materials, 2021, doi:10.3390/ma14195611_

Round 1

Reviewer 1 Report

My overall impressions about the general quality and the scientific soundness of the paper are positive. The authors provided here a detailed characterization of the exciting and important phenomenon. In my opinion, the presented results are of good quality and well-described. Therefore, the paper deserves to be considered for publication. Also, all tables, figures, and formulas are provided with good quality. The language is acceptable, and only some typing errors appear within the text. Before the acceptance, the author should follow the minor comments below:

  1. In line 144, the authors say that they assume constant temperature. However, it seems that in the grinding process, the rise of the temperature can occur. Did you check somehow the possible presence of such an increase?
  2. All variables that appear in the text should be in italic style. Also, on page 4, lines 159 and 160, the subscripts are missing in variables v and f.
  3. In line 174, there is softer instead software.
  4. In lines 172-173, the authors assumed the clearance between the wall and the grinding wheel to be 0.01mm. Have you checked other values, also how can you prove such a choice? It seems the choice of this clearance influences the value of the disturbance pressure.
  5. R’ – missing description

In summary, I recommend the minor revision of the paper, according to the comments above.

Author Response

We have revised the manuscript according to the reviewer's comment and marked in red.

Point 1: In line 144, the authors say that they assume constant temperature. However, it seems that in the grinding process, the rise of the temperature can occur. Did you check somehow the possible presence of such an increase?

Response 1: Thank you for your suggestion. In fact, the temperature in the grinding zone will increase, and we have investigated this, as presented in literature [1]. The effect of temperature rise on the surrounding liquid can be ignored. Therefore, we assumed that the temperature and density of the surrounding liquid are constant.

[1] Chen F., Li G.X., Zhao B., Bie W.B. Thermomechanical coupling effect on characteristics of oxide film during ultrasonic vibration-assisted ELID grinding ZTA ceramics[J]. Chinese Journal of Aeronautics, 2021, 34(6): 125-140.

Point 2: All variables that appear in the text should be in italic style. Also, on page 4, lines 159 and 160, the subscripts are missing in variables v and f.

Response 2: We have revised the style of the variables as follows:

wheredenotes the ultrasonic amplitude, denotes the axial feed rate of grinding wheel, denotes the ultrasonic frequency, and  denotes the feed rate of workbench.

And other variables' style has been modified, as detailed in the revised manuscript.

Point 3: In line 174, there is softer instead software.

Response 3: We have replaced softer with software.

Point 4: In lines 172-173, the authors assumed the clearance between the wall and the grinding wheel to be 0.01mm. Have you checked other values, also how can you prove such a choice? It seems the choice of this clearance influences the value of the disturbance pressure.

Response 4: According to the comment, we have added the relative content as follows:

In this paper, the diameter of the grinding wheel was 25 mm, and the grain size is 3.5 to 40μm. In addition, the ultrasonic frequency ranged from 20 to 35 kHz, and the ultrasonic amplitude is approximated 10μm. Since the clearance between the grinding wheel and workpiece during processing is close to the prominent height of the grinding, and the effective grinding size is only 1/3 of the grinding particle size. Besides, considering the wear of grinding particle, the clearance between the wall and the grinding wheel was assumed to be 0.01mm.

Point 5: R’ – missing description

Response 5: is the liquid radial speed at the bubble boundary

Author Response

We have revised the manuscript according to the reviewer's comment and marked in blue.

Point 1:It is better that the title should not have any abbreviations.

Response 1: According to the comment, we have revised the title as “Cavitation effect in ultrasonic-assisted electrolytic in-process dressing grinding of nanocomposite ceramics”.

Point 2: The proficiency of the language needs more improvement in the manuscript. There are many long sentences, that make the reader lose his focus. For example. 40-43.

Response 2: Thank you for your suggestion. We have modified the long sentences in the previous manuscript. For example, “Therefore, alternative processing technologies have been introduced, such as the elastic emission machining (EMM) [4], pre-stressed processing [5], rotary ultrasonic machining (RUM) [6], ELID ultra-precision grinding [7], magnetic polishing technology [8], ultrasonic-assisted grinding [9], and UA-ELID grinding [10]”.

Point 3: The subscript of all ceramics needs to be revised. For example, in L32, 36…. Al2O3 should be Al2O3.

Response 3: According to the comment, we have revised the subscript of all ceramics, as shown in the revised manuscript.  

Point 4: I recommend collecting all abbreviations in a nomenclature table as attached table.

Response 4: Thank you for your suggestion. We have collected all abbreviations in a nomenclature table as follows.

The abbreviations were added before “References” in the revised manuscript.

Reviewer 3 Report

Article by title: "Cavitation effect in ultrasonic-assisted ELID grinding of nanocomposite ceramics" Authors: Guangxi Li , Fan Chen * , Wenbo Bie , Bo Zhao , Zongxia Fu , Xiaobo Wang, is good written.   The paper was presented ultrasonic-assisted electrolytic in-process dressing (UA-ELID) grinding. UA-ELID  is a promising technology that uses a metal-bonded diamond grinding wheel to achieve a mirror surface finish on hard and brittle materials.

Manuscript is prepared properly in terms of substance and form. I propose to submit the article for publication in the given form.

Author Response

Thank you very much for the reviewer's approval.